

# Economic damage and spill-overs from a tropical cyclone
Manfred Lenzen[1], Arunima Malik[1], Steven Kenway[2], Peter Daniels[3], Ka Leung Lam[2], Arne
Geschke[1]
[1]ISA, School of Physics A28, The University of Sydney, NSW, 2006, Australia.
[2]School of Chemical Engineering, The University of Queensland, St Lucia, 4072, Australia.
[3]School of Environment, Griffith University, Brisbane, 4222, Australia.
*Correspondence to*: Arunima Malik (arunima.malik@sydney.edu.au)



**Abstract** – Tropical cyclones cause widespread damage in specific regions as a result of high winds, and flooding. Direct
impacts on commercial property and infrastructure can lead to production shortfalls. Further losses can occur if business
continuity is lost through disrupted supply of intermediate inputs from, or distribution to, other businesses. Given that
producers in modern economies are strongly interconnected, initially localised production shortfalls can ripple through
entire supply-chain networks and severely affect the regional and wider national economy. In this paper, we use a
comprehensive, highly disaggregated, and recent multi-region input-output framework to analyse the impacts of Tropical
Cyclone Debbie. In particular, we show how industries and regions that were not directly affected by storm and flood
damage suffered significant job and income losses. Our results indicate that the disaster resulted in the direct loss of
about 7,000 full-time equivalent jobs and 2 billion AUD of value added, and an additional indirect loss of 5,000 jobs and
1 billion AUD of value added. We are able to conduct this assessment so rapidly due to the timely data provision and
collaborative environment facilitated by the Australian Industrial Ecology Virtual Laboratory (IELab).
**Keywords:** Tropical cyclone, economic damage, spill-over, input-output analysis, hurricane, typhoon



## 1. Introduction

On Tuesday 28 March 2017, Severe Tropical Cyclone Debbie made landfall at Airlie Beach, in North Queensland, Australia. As a Category 4 system (equivalent to a major Hurricane or a Typhoon), it hit coastal communities with torrential rain and wind gusts up to 265 km/h, destroying or damaging homes, businesses, crops and infrastructure and, tragically, led to 12 fatalities (Queensland Government, 2017). The initial impact fell mainly on the iconic coral reef region of the Whitsunday Coast, and the surrounding communities including Bowen and Proserpine. Within 24 hours, Debbie was approximately 250 km inland, and had degenerated into a high-rainfall low-pressure system. The system progressively tracked over 1,000 km south, where it moved back out to sea around the Queensland-New South Wales Border on 31 March after significant flooding across the region. Rainfall of 150-250 mm was recorded regionally, with peaks of 400-1,000 mm, swamping remote rural, coastal and urban communities. More than a week later, widespread flooding was still being felt (Queensland Government, 2017).

Dubbed the "Lazy Cyclone", Debbie moved at under 6 km/h at times, leading to atypically high levels of social, economic, and environmental destruction. Over 63,000 emergency calls were made, and over 50,000 insurance claims subsequently lodged (Queensland Government, 2017). Particular impact was felt in the farming, mining and tourism industries in the north, and by flooded businesses in the south. Annual and perennial crops and trees were destroyed, export-oriented coal mines closed, and tourism heavily impacted. Roads, rail systems and bridges were damaged or destroyed, along with community halls, airfields, tele-communications and other systems. All schools and many businesses were temporarily closed. The Australian government responded at all levels including federal military deployment of air, sea and land support, Queensland Police, Fire and Emergency, and State Emergency Systems.

Past tropical cyclones, in Australia and elsewhere, have disrupted food supply – for example in Madagascar (Gafilo, 2004), Vanuatu (Pam, 2015), and Fiji (Winston, 2016). If frequent enough, cyclones could affect food security even on a large continent like Australia, as the cyclone-prone area of coastal Queensland produces up to three quarters of Australia's perishable vegetables. In 2006, Cyclone Larry wiped out 80% of Australia's banana crop and left thousands of people without work (Staff, 2017). In 2011, Cyclone Yasi lead to a shortage of bananas in Australia (Brown, 2017).

Economic damage caused by Debbie has been estimated to include $1.5 billion (AUD) in lost coal sales, and approximately $AUD 0.5 billion in agriculture, with sugar cane and winter horticulture supplies to southern Australia particularly influenced. Infrastructure damage has been estimated at over $1 billion (Queensland Government, 2017). Flood damage to business and trade was also significant in northern New South Wales. Debbie also caused temporary shortages to water and energy supplies (Parnell, 2017), damaged information technology infrastructure, and led to price increases for tomatoes, capsicums, eggplants, and other vegetables (Hatch, 2017), affecting winter vegetable supply for Sydney and Melbourne. Insurance claims of over $300 million were lodged.



Developing and testing methods for assessing economic consequences of natural disasters is of growing importance,
because of the increasing frequency of extreme weather events such as tropical cyclones that can be attributed to climate
change (Mendelsohn et al, 2012). In our case study, this significance is reinforced given the importance of Northern
Australia in plans for the nation's ongoing economic development, notably in mining and agriculture (Regional Institute
of Australia, 2013).
In this paper, we determine the full supply-chain impacts of Tropical Cyclone Debbie, using highly disaggregated multi-
regional input-output (MRIO) tools developed within the new Australian Industrial Ecology Virtual Laboratory (IELab;
described in Section 3.4). More specifically, we apply the Australian IELab to construct a customised sub-national MRIO
table for Australia with extensive detail on regions directly affected by the cyclone. Our approach incorporates a number
of new capabilities. First, we are able to examine spill-overs across 19 regions and 34 industry sectors; in other words, we
identify the consequences of the cyclone not only for the directly affected regions and industry sectors, but for the wider
Australian economy. This highlights the innovative strength of the Australian IELab, which offers unprecedented spatial
resolution, hence allows for a comprehensive assessment of the direct as well as indirect supply-chain effects of the
disaster, and the ability to trace spill-overs. In addition, the IELab offers sophisticated tools which to our knowledge have
so far not been applied to disaster analysis: For example, Production Layer Decomposition is able to pinpoint the
sequence of indirect impacts rippling across the regional supply-chain network. One further advanced capability is the in-
built data updating functionality in the IELab, allowing for the provision of recent economic and social data-sets,
enabling the timely and cost-effective analysis of disasters to support expeditious decision-making. Finally, the IELab
also offers data-sets and analytical tools for assessing the local/regional effects in terms of a range of physical indicators,
such as carbon dioxide emissions, water use, energy use and waste, to name a few. Whilst such an assessment is beyond
the scope of this study, this is surely an area of research that warrants further investigation.
This article is structured as follows: Section 2 provides a review of relevant prior work and the state of knowledge in
input-output (IO) based disaster analysis. Section 3 describes the methodology underlying the disaster analysis
undertaken using input-output modelling. In particular, we build on prior work (Schulte in den Bäumen et al., 2015) and
present an innovative approach for estimating infrastructure damages resulting from the disaster. We present the results
and a discussion of key findings in Section 4, followed by conclusions in Section 5.

108       **2.   Input-output based disaster analysis**

Input-output (IO) analysis studies feature a sub-stream dealing with disaster analysis. Okuyama (2007) provides a
comprehensive review of the use of input-output analysis for economic analysis of disasters. Quantitative disaster
analysis is needed for understanding the impacts of a disaster, for driving effective disaster response, for informing
disaster risk reduction and adaptation efforts, and for pre-emptive planning and decision-making (Lesk et al., 2016;
Temmerman et al., 2013; Prideaux, 2004; Cannon, 1993). It is intuitively clear that a disaster results in direct losses in the
form of infrastructure damages, and indirect higher-order effects in the form of subsequent losses in business activity





(Rose, 2004). The ability of IO analysis in capturing the upstream interconnected supply chains of an industry or region
affected by a disaster makes it an ideal tool for assessing the full scope of impacts of a disaster event. In addition to IO
analysis, computable general equilibrium (CGE) models, econometric models and social accounting matrices (SAM) are
alternative modelling frameworks for estimating the indirect higher-order effects of a disaster (Okuyama, 2007;
Okuyama and Santos, 2014; Tsuchiya et al., 2007; Rose and Liao, 2005; Rose and Guha, 2004; Cole, 1995; Guimaraes et
al., 1993; Koks et al., 2016; Koks and Thissen, 2016). A discussion of these models is beyond the scope of this study and
we focus on IO analysis, in particular the post-disaster consumption possibilities, and possible spill-overs (explained
further below). IO modelling has been applied to many disasters such as earthquakes in Japan (Okuyama, 2014, 2004),
floods in Germany (Schulte in den Bäumen et al., 2015) and London (Li et al., 2013), terrorism (Lian and Haimes, 2006;
Santos and Haimes, 2004; Rose, 2009), hurricanes (Hallegatte, 2008) and blackouts (Anderson et al., 2007) in the USA,
and  diseases and epidemics (Santos et al., 2013; Santos et al., 2009), to name a few.
Prior research on disaster impact analysis, based on IO analysis, has sought out ways of improving the existing IO model
by extending the standard framework to include temporal and spatial scales (Okuyama, 2007): For example, Donaghy et
al. (2007) propose a flexible framework for incorporating short- and long-time frames using the regional econometric
input-output model (REIM), and Yamano et al. (2007) apply a regional disaggregation method to a multi-regional input-
output (MRIO) model to estimate higher-order effects according to specific districts. Furthermore, a so-called
inoperability index within the inoperability input-output model (IIOM) has been proposed as a way of assessing the effect
of a disaster or initial perturbation on interconnected systems (Haimes et al., 2005). Both the static and the dynamic
versions of IIOM have been applied to the case of terrorism for assessing the economic losses resulting from
interdependent complex systems (Santos and Haimes, 2004; Lian and Haimes, 2006). Using the dynamic version of
IIOM, it is possible to assess recovery times and also to identify systems and economic sectors that are most critical for
guiding the recovery process (Haimes et al., 2005).
One particular type of disaster IO analysis, proposed by Steenge and Bočkarjova (2007) aims at investigating post-
disaster consumption possibilities as a consequence of production shortfalls resulting from a disaster. Such an assessment
has been applied, for example to widespread flooding in Germany (Schulte in den Bäumen et al., 2015) and electricity
blackouts from possible severe space weather events (Schulte in den Bäumen et al., 2014). Here, we undertake an
estimation of post-disaster consumption possibilities, and subsequent losses in employment and economic value added
resulting from the 2017 Tropical Cyclone Debbie in Australia. To this end, we use the Australian IELab to construct a
customised sub-national MRIO table for Australia with extensive detail on regions directly affected by the cyclone. In
particular, and this is the novelty of our research, we examine detailed *regional and sectoral spill-overs* including the
consequences of this cyclone not only for directly affected regions and industry sectors, but also for the wider national
economy.




## 3. Methods

In this work, we use multi-region economic input-output (MRIO) analysis in order to investigate the economy-wide repercussions of the damage wrought by Tropical Cyclone Debbie in the North Queensland region of Australia. Input-output (IO) analysis (IO) is an economic technique conceived in the 1930s by Nobel Prize Laureate Wassily Leontief (Leontief, 1936). IO analysis is able to interrogate economic data on inter-industry transactions, final consumption and value added, in order to trace economic activity rippling throughout complex supply-chain networks and unveil both immediate and indirect impacts of systemic shocks (Leontief, 1966). Over the past seventy years, IO analysis has been used extensively for a wide range of public policy and scientific research questions (Rose and Miernyk, 1989). Over the past two decades, IO analysis has experienced a surge in applications, especially on carbon footprints (Wiedmann, 2009) and global value chains (Timmer et al., 2014), and in the disciplines of life-cycle assessment (Suh and Nakamura, 2007) and industrial ecology (Suh, 2009).

### 3.1 Input-output disaster analysis

A specific stream of IO analysis is disaster analysis (Okuyama, 2014, 2007), focused upon input-output databases employed to explore how an economy can be affected by a sudden slowdown or shutdown of individual industries. Since we are primarily interested in post-disaster consumption possibilities and ensuing employment and value-added loss, we utilise the approach by Steenge and Bočkarjova (2007). In essence, a disaster reduces total economic output $\mathbf{x}_0$ of industry sectors $1,\ldots,N$ to levels

$$\tilde{\mathbf{x}} = (\mathbf{I} - \boldsymbol{\Gamma})\mathbf{x}_0, \tag{1}$$

where $\boldsymbol{\Gamma}$ is a diagonal matrix of fractions describing sectoral production losses as a direct consequence of the disaster, and $\mathbf{I}$ is an identity matrix with the same dimensions as $\boldsymbol{\Gamma}$. Post-disaster consumption possibilities $\mathbf{y}_1$ are then the solution of the linear problem

$$\max(\mathbf{1}\mathbf{y}_1) \text{ s.t. } \text{i) } \mathbf{y}_1 = (\mathbf{I} - \mathbf{A})\mathbf{x}_1 \text{ , } \text{ ii) } \mathbf{x}_1 \le \tilde{\mathbf{x}} \text{ , and } \text{iii) } \mathbf{y}_1 \ge 0 \text{ ,} \tag{2}$$

where $\mathbf{1} = \underbrace{[1,1,\ldots,1]}_{N}$ is a summation operator, $\mathbf{A} = \mathbf{T}\widehat{\mathbf{x}_0}^{-1}$ is a matrix of input coefficients, $\mathbf{T}$ is the intermediate transactions matrix, the '^' (hat) symbol denotes vector diagonalisation, and $\mathbf{x}_1$ is post-disaster total economic output. Constraint i) in Equation (2) is the standard fundamental input-output accounting relationship stating that in every economy intermediate demand $\mathbf{T}$ and final demand $\mathbf{y}$ sum up to total output $\mathbf{x}$. Constraint ii) states that in the short term, post-disaster total output is limited by pre-disaster total output minus disaster-induced losses. Constraint iii) ensures that final demand is strictly positive. Note that positive offsetting effects can result from natural disasters. For example, the replacement or repairs to damaged buildings and infrastructure, or any other demand for commodities required especially





for post-disaster recovery, are likely to create additional employment and value added and may embody updated
technology. In addition, above-average rainfall may have been beneficial for pastures and water supply, and increased
freshwater run-off and turbidity could have increased catches of prawn trawling. However, these effects are not
accounted for in our study. Supplementary Information S2 offers further details on the approach suggested by Steenge
and Bočkarjova (2007) .

3.2  Disaster impact on value added and employment

A disaster-induced transition to lower consumption levels $\mathbf{y}_1 = \mathbf{y}_0 - \Delta\mathbf{y}$ has implications for the state of regional
economies, as it causes losses in value added and employment

$\Delta Q = \mathbf{q}\Delta\mathbf{x} = \mathbf{q}(\mathbf{I} - \mathbf{A})^{-1}\Delta\mathbf{y}$ ,              (3)

where $\mathbf{q}$ holds value-added and employment coefficients. The sequence of these losses can be enumerated by carrying
out a *production layer decomposition*, that is by unravelling the inverse in Eq. (3) into an infinite series (see (Waugh,
1950)) as

$\Delta Q = \mathbf{q}\Delta\mathbf{y} + \mathbf{qA}\Delta\mathbf{y} + \mathbf{qA}^2\,\Delta\mathbf{y} + +\mathbf{qA}^3\,\Delta\mathbf{y} + \cdots = \sum_{n=0}^{\infty}\mathbf{qA}^n\,\Delta\mathbf{y}$ ,     (4)

where the term $\mathbf{q}\Delta\mathbf{y}$ represent the immediate job and value-added losses in the regions directly hit by the cyclone, $\mathbf{qA}\Delta\mathbf{y}$
describes $1^{\text{st}}$-order losses fielded by suppliers of cyclone-affected producers, $\mathbf{qA}^2\,\Delta\mathbf{y}$ $2^{\text{nd}}$-order losses for suppliers of
suppliers, and so on for subsequent upstream production layers. $1^{\text{st}}$- and higher-order upstream losses can in principle
occur anywhere in Australia, depending on the reach of the supply-chain network of local northern Queensland
producers.


3.3  Case study: Tropical Cyclone Debbie

In order to quantify indirect economic impacts of Cyclone Debbie, we first constructed a 19-region, by 34-sector input-
output model of Australia, with particular regional detail for the regions close to disaster centres, ie 10 subregions of
Queensland as well as northern New South Wales (see also Fig 1). The compilation of this table and underlying data are
outlined in Section 3.4.

3.3.1     Reduction in industry output, and creation of the gamma matrix

In order to estimate indirect consequences of Cyclone Debbie we further developed the method of Schulte in den
Bäumen et al. (2015) and created the so-called gamma matrix (Eq. (1)): a diagonal matrix of "gamma fractions" $\Gamma_j$ (See



Eq. (1)) of production possibilities lost due to the cyclone (19 * 34 region-sector pairs, square). We determined the
relative reductions in industry output by (a) sourcing public information on actual or estimated financial damages and (b)
dividing these by gross output taken from our MRIO table. Information on damages included (a) the reduction of total
industry output (in 2017 compared to 2016), plus (b) an annualised value of infrastructure damage, explained below. A
value of $\Gamma_j = 0.1$ indicates a 10% loss of production value (including related infrastructure costs) from 2016 to 2017.
Information on the direct damages by the cyclone was sourced from a range of published government reports, through
personal enquiries to government, from government statements and websites, and from media and industry reports. Table
1 provides a summary of main impacts; further details and related data sources are provided in SI2.2, including a
summary of infrastructure damage caused by the cyclone shown in Table SI2.3.
Damages were only considered where we could find empirical monetary information. It is likely that other negative
impacts, and also some positive impacts may have occurred in other regions. For example, increased water capture and
storage for irrigation, the replacement of outdated technology with new infrastructure, or jobs created through
construction required as a consequence of the cyclone damage. As no data were available for quantifying such
repercussions, we did not consider such impacts.
3.3.2    Estimation of infrastructure damage
Infrastructure damage from the cyclone in the state of Queensland was estimated at well over a billion dollars
(Queensland Government, 2017). The localities of Mackay and Fitzroy had bridges, roads, airport, community
infrastructure, water and wastewater treatment plants damaged or destroyed. High impacts were also noted in Richmond-
Tweed (from significant flooding), and in Brisbane (over seven bridges damaged, significant degradation of at least 350
local roads and 200 major culverts etc), as well as northern Queensland (see Supplementary Information S3 for details).
As an innovation on the work of Schulte in den Bäumen et al. (2015), we estimated infrastructure damage and its
attribution to sectors of the economy using an "infrastructure gamma fraction". This was undertaken by adding to the
production possibilities loss matrix $\Gamma$: In addition to the conventional current output losses, we estimated losses caused
by damage to infrastructure such as roads. We attributed such infrastructure damage values to sectors of the economy by
annualising the dollar value impact of infrastructure as a fraction of Gross Operating Surplus (GOS), derived from the
MRIO database. We assumed that the damaged industries required a fraction of this surplus to replace capital (i.e. the
infrastructure) over a 25-year time-frame. The main infrastructure impacts of the cyclone were borne in sectors such as
electricity, gas, water, trade, accommodation, cafes, restaurants, road transport, rail and pipeline transport, other
transport, and communication services. As an example, 50 $m of infrastructure damage, spread over 25 years, equal a 2
$m cost deductible from GOS. A similar, more generalised approach has been outlined by Hallegatte (2008). The total
production loss coefficients (gamma fraction) in $\Gamma$ were calculated by adding the (a) current output losses (See 3.3.1) and
the losses induced by infrastructure damage (Table 2).



Table 1: Summary of major direct impacts (see Supplementary Information SI2 for details and sourcing).

| Aspect | Region | Industries | Example impact |
|---|---|---|---|
| Coal exports | All QLD | Coal, oil and gas | Coal exports may have taken a $1.5 billion hit from Cyclone Debbie as more than 22 mines were forced to halt production while roads and ports were shut. |
| Sugar Cane | QLD- Mackay | Sugar cane growing | Damage to Queensland's sugar industry is expected to cost A$150 million (US$114.4 million). The majority of these costs lie in Proserpine and Mackay. |
| Vegetables | QLD-Mackay | Other agriculture | The Queensland Farmers Federation (QFF) said early figures show actual crop damage to Bowen's vegetable industry is about $100 million, accounting for about 20 per cent of the season's crop. |
| Vegetables | NSW Richmond & Tweed | Other agriculture | Lost nut production would have been worth about $8 million. |
| Agriculture, grains and sugarcane | All QLD regions and NSW Richmond & Tweed. | Grains Other agriculture Sugar cane growing | The National Farmers' Federation has cited industry groups estimating damage to crops of up to $1 billion. |
| Business | NSW Richmond & Tweed. | Accommodation, Cafes, and Restaurants, Trade | 50 to 80 per cent of these businesses [will not reopen] in the community of 50,000 people. |
| Dairy | QLD - Brisbane | Dairy cattle and pigs | It is anticipated that the cost to the farming industry in South East Queensland will be in excess of $6 million. |
| Infrastructure | All QLD | Multiple industries | The cost of recovery would 'be in the billions' of dollars, with roads, bridges, crops, homes and schools all needing serious repairs. |
| Insurance | All 19 regions (with most focus on QLD and Northern NSW) | Multiple Industries | Insurance losses $A306 million. Over a $1bill in insurance claims. |
| Fatalities | - | - | 12 Fatalities. |
| Evacuation costs | - | - | 25,000 residents evacuated in Mackay, and 55,000 in Bowen. |
| Schools | - | - | 400 schools closed. |
| Airflights | - | - | Flights cancelled Townsville from March 27. Virgin Airlines losses in the 3 months to March $62.3 million was impacted by Cyclone Debbie. |
| Rail | - | - | QLD Rail suspended trains between Rocky and Townsville NQ Bulk Ports closed at Mackay, Abbot Point and Hay Point. |
| Emergency workers | - | - | 1,000 emergency workers deployed, 200 Energex workers. |
| Defence forces | - | - | 1,200 personnel deployed. |


Table 2 – Entries of the $\Gamma$ matrix (production possibilities lost) including (a) industry output and (b) infrastructure costs annualised over 25 years. Note that a fraction of 0.1 means a 10% reduction in reduced production (between 2016 and 2017) including both lost productivity plus a share of cost relating to infrastructure-damage (annualised over 25 years).

| | Rest of NSW | NSW-Richmond-Tweed | VIC | QLD-Brisbane | QLD-Wide Bay Burnett | QLD-Darling Downs | QLD-South West | QLD-Fitzroy | QLD-Central West | QLD-Mackay | QLD-Northern | QLD-Far North | QLD-North West | SA | WA | TAS | ACT | NT |
|---|---|---|---|---|---|---|---|---|---|---|---|---|---|---|---|---|---|---|
| 1 Sheep | 0 | 0 | 0 | 0 | 0 | 0 | 0 | 0 | 0 | 0 | 0 | 0 | 0 | 0 | 0 | 0 | 0 | 0 |
| 2 Grains | 0 | 0 | 0 | 0 | 0 | 0 | 0 | 0 | 0 | 0 | 0 | 0 | 0 | 0 | 0 | 0 | 0 | 0 |
| 3 Beef cattle | 0 | 0 | 0 | 0 | 0 | 0 | 0 | 0 | 0 | 0 | 0 | 0 | 0 | 0 | 0 | 0 | 0 | 0 |
| 4 Dairy cattle and pigs | 0 | 0 | 0 | 0.110 | 0 | 0 | 0 | 0 | 0 | 0 | 0 | 0 | 0 | 0 | 0 | 0 | 0 | 0 |
| 5 Other agriculture | 0 | 0.070 | 0 | 0 | 0 | 0 | 0 | 0 | 0.314 | 0.530 | 0 | 0 | 0 | 0 | 0 | 0 | 0 | 0 |
| 6 Sugar cane growing | 0 | 0 | 0 | 0 | 0 | 0 | 0 | 0.377 | 0 | 0.264 | 0.111 | 0 | 0 | 0 | 0 | 0 | 0 | 0 |
| 7 Forestry and fishing | 0 | 0 | 0 | 0 | 0 | 0 | 0 | 0 | 0 | 0 | 0 | 0 | 0 | 0 | 0 | 0 | 0 | 0 |
| 8 Coal, oil and gas | 0 | 0 | 0 | 0 | 0 | 0 | 0 | 0.168 | 0 | 0.176 | 0.246 | 0 | 0 | 0 | 0 | 0 | 0 | 0 |
| 9 Non-ferrous metal ores | 0 | 0 | 0 | 0 | 0 | 0 | 0 | 0 | 0 | 0 | 0 | 0 | 0 | 0 | 0 | 0 | 0 | 0 |
| 10 Other mining | 0 | 0 | 0 | 0 | 0 | 0 | 0 | 0 | 0 | 0 | 0 | 0 | 0 | 0 | 0 | 0 | 0 | 0 |
| 11 Food manufacturing | 0 | 0 | 0 | 0 | 0 | 0 | 0 | 0 | 0 | 0 | 0 | 0 | 0 | 0 | 0 | 0 | 0 | 0 |
| 12 Textiles, clothing and footwear | 0 | 0 | 0 | 0 | 0 | 0 | 0 | 0 | 0 | 0 | 0 | 0 | 0 | 0 | 0 | 0 | 0 | 0 |
| 13 Wood and paper manufacturing | 0 | 0 | 0 | 0 | 0 | 0 | 0 | 0 | 0 | 0 | 0 | 0 | 0 | 0 | 0 | 0 | 0 | 0 |
| 14 Chemicals, petroleum and coal products | 0 | 0 | 0 | 0 | 0 | 0 | 0 | 0 | 0 | 0 | 0 | 0 | 0 | 0 | 0 | 0 | 0 | 0 |
| 15 Non-metallic mineral products | 0 | 0 | 0 | 0 | 0 | 0 | 0 | 0 | 0 | 0 | 0 | 0 | 0 | 0 | 0 | 0 | 0 | 0 |
| 16 Metals; metal products | 0 | 0 | 0 | 0 | 0 | 0 | 0 | 0 | 0 | 0 | 0 | 0 | 0 | 0 | 0 | 0 | 0 | 0 |
| 17 Machinery appliances and equipment | 0 | 0 | 0 | 0 | 0 | 0 | 0 | 0 | 0 | 0 | 0 | 0 | 0 | 0 | 0 | 0 | 0 | 0 |
| 18 Miscellaneous manufacturing | 0 | 0 | 0 | 0 | 0 | 0 | 0 | 0 | 0 | 0 | 0 | 0 | 0 | 0 | 0 | 0 | 0 | 0 |
| 19 Electricity supply, gas and water | 0 | 0.003 | 0 | 0.001 | 0 | 0 | 0 | 0.002 | 0 | 0.019 | 0.001 | 0 | 0 | 0 | 0 | 0 | 0 | 0 |
| 20 Residential building construction | 0 | 0 | 0 | 0 | 0 | 0 | 0 | 0.053 | 0 | 0.055 | 0.042 | 0 | 0 | 0 | 0 | 0 | 0 | 0 |
| 21 Other construction | 0 | 0 | 0 | 0 | 0 | 0 | 0 | 0 | 0 | 0 | 0 | 0 | 0 | 0 | 0 | 0 | 0 | 0 |
| 22 Trade | 0 | 0.072 | 0 | 0.008 | 0 | 0 | 0 | 0.039 | 0 | 0.036 | 0.021 | 0 | 0 | 0 | 0 | 0 | 0 | 0 |
| 23 Accommodation, cafes and restaurants | 0 | 0.225 | 0 | 0.005 | 0 | 0 | 0 | 0.005 | 0 | 0.100 | 0.005 | 0 | 0 | 0 | 0 | 0 | 0 | 0 |
| 24 Road transport | 0 | 0.019 | 0 | 0.002 | 0 | 0 | 0 | 0.058 | 0 | 0.095 | 0.010 | 0 | 0 | 0 | 0 | 0 | 0 | 0 |
| 25 Rail and pipeline transport | 0 | 0 | 0 | 0 | 0 | 0 | 0 | 0.013 | 0 | 0.0 | 0 | 0 | 0 | 0 | 0 | 0 | 0 | 0 |
| 26 Other transport | 0 | 0 | 0 | 0 | 0 | 0 | 0 | 0.007 | 0 | 0.0 | 0 | 0 | 0 | 0 | 0 | 0 | 0 | 0 |
| 27 Communication services | 0 | 0 | 0 | 0 | 0 | 0 | 0 | 0.013 | 0 | 0.035 | 0.001 | 0 | 0 | 0 | 0 | 0 | 0 | 0 |
| 28 Finance, property and business services | 0 | 0 | 0 | 0 | 0 | 0 | 0 | 0 | 0 | 0 | 0 | 0 | 0 | 0 | 0 | 0 | 0 | 0 |
| 29 Ownership of dwellings | 0 | 0 | 0 | 0 | 0 | 0 | 0 | 0 | 0 | 0 | 0 | 0 | 0 | 0 | 0 | 0 | 0 | 0 |
| 30 Government administration and defence | 0 | 0 | 0 | 0 | 0 | 0 | 0 | 0 | 0 | 0 | 0 | 0 | 0 | 0 | 0 | 0 | 0 | 0 |
| 31 Education | 0 | 0 | 0 | 0 | 0 | 0 | 0 | 0 | 0 | 0 | 0 | 0 | 0 | 0 | 0 | 0 | 0 | 0 |
| 32 Health and community services | 0 | 0.010 | 0 | 0 | 0 | 0 | 0 | 0 | 0 | 0 | 0 | 0 | 0 | 0 | 0 | 0 | 0 | 0 |
| 33 Cultural and recreational services | 0 | 0.018 | 0 | 0.002 | 0 | 0 | 0 | 0.078 | 0 | 0.504 | 0.008 | 0 | 0 | 0 | 0 | 0 | 0 | 0 |
| 34 Personal and other services | 0 | 0 | 0 | 0 | 0 | 0 | 0 | 0.135 | 0 | 0 | 0 | 0 | 0 | 0 | 0 | 0 | 0 | 0 |

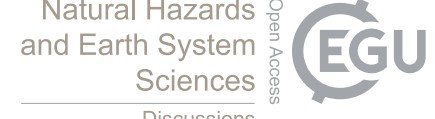



### 3.4 Data

Data for the production recipe **A** and initial total output $\mathbf{x}_0$ were taken from the Australian Industrial Ecology Virtual Laboratory (IELab; (Lenzen et al., 2014). The Australian Industrial Ecology Virtual Laboratory (IELab) is a cloud-computing environment that allows the construction of customised input-output (IO) databases. IO tables document the flow of money between various industries in an economy – national input-output tables present national data on intra- and inter-industry transactions between industries in a national economy, whereas multi-regional input-output (MRIO) tables harbour detailed data on trade between two different regions (Tukker and Dietzenbacher, 2013); see (Leontief, 1953) for an account of MRIO theory). MRIO tables can either be global or sub-national. Global tables feature more than one country, and provide detailed data on international trade between countries, whereas sub-national MRIO tables provide detailed trade data for regions within one country. These tables have been extensively used for undertaking environmental, social and economic footprint assessments (Lenzen et al., 2012; Alsamawi et al., 2014; Oita et al., 2016; Wiedmann et al., 2013; Simas et al., 2014; Hertwich and Peters, 2009). Coupling of economic MRIO data with so-called physical accounts, as conceived by Nobel Prize winner Wassily Leontief in the 1970s, allows for the enumeration of direct as well as indirect supply-chain impacts (Leontief, 1966, 1970).

The IELab is capable of generating *multi-region input-output* (MRIO) databases, where industry sectors can be distinguished for a number of Australian regions. Users are able to choose from a set of 2214 statistical areas (Level 2; (ABS, 2016a)) to delineate MRIO regions with their specific research question in mind. The regional and sectoral flexibility of the IELab (see (Lenzen et al., 2017)) was exploited by generating a regional partition of Australia that is more detailed around the regions where the cyclone caused most of its damage (Queensland and Northern New South Wales), and less detailed elsewhere (Fig. 1). As a sectoral breakdown we use the 34-sector industry classification from the Queensland regional input-output database ((OGS, 2004); see Supplementary Information S1).

A number of national, state and region-specific data sources were used for constructing the MRIO database used in this work. These are the income, expenditure and product accounts (ABS, 2016c) the input-output tables (ABS, 2017, 2016g) for the national level; the state accounts (ABS, 2016f) and the Queensland input-output tables (OGS, 2002) for the state level; and the household expenditure survey (ABS, 2011), Queensland regional input-output tables (OGS, 2004), the business register (ABS, 2016e), the census (ABS, 2012) and the agricultural commodities survey (ABS, 2016d) for the regional level. Detailed regional employment data were taken from the labour force survey (ABS, 2016b).

### 3.4.1 Superior economic data

In order to be meaningful, any regional input-output analysis needs to be supported by specific regional data (see an IELab-based analysis of Western Australia by Lenzen et al. (2017)). We therefore sourced superior economic data to update the IO data for sub-regions and sectors most affected by Cyclone Debbie, with the most recent financial and economic information available. In particular, data were sought covering value of production, total output, salaries paid,





gross operating values, regional export, turnover, and regional economic productivity (Table 3). Key sources of
information included accounts published by the Australian Bureau of Statistics (ABS), e.g. covering the gross value of
agriculture and manufacturing sales and wages. Grey literature including regional economic studies, value of production
accounts kept by State agencies, and Treasury investigations also provided important data within which to constrain the
reconciliation of our MRIO base table. Collectively, the superior data added more than 800 data points of information by
which the accuracy of MRIO entries for Queensland and Northern NSW was improved.





Table 3 Summary of superior economic data used as constraints in compilation of the MRIO.

| Data aspect | Region | Sector/s | Years | Example data | Reference |
|---|---|---|---|---|---|
| GRP | All Queensland sub-regions. | All | 2010-11 | GRP Mackay 2011 = $22 billion | (Queensland Treasury and Trade, 2013) |
| GRP - Richmond Tweed | NSW - Richmond & Tweed | All | 2011-12 | GRP > $8.5 billion | (Wilkinson, 2014) |
| Manufacturing sales & service income, wages and salaries, employment | 10 QLD regions and NSW-Richmond &Tweed | Food product manufacturing and all other manufacturing | 2006-07 is latest | Food product manufacturing in Mackay = $1,051 million in 2007. | (ABS, 2008) |
| Manufacturing sales & service income, wages and salaries, employment | QLD – all regions | Food product manufacturing and all other manufacturing | 2010-11 to 2014-15 | Food product manufacturing in QLD = $20,131 million in 2015. | (Australian Bureau of Statistics, 2016a), (Australian Bureau of Statistics, 2017) |
| Coal | QLD – all regions | Coal, oil and gas | 2015-16 | Production value by SA4** area, eg $19.437 billion sales for 2015-16 calendar year with $12.234 billion in SA4 Mackay; and $6.170 billion in Fitzroy. | (Keir, 2017) |
| Import and export of horticulture products | QLD – all regions | Part of Other agriculture | 2014-15 | $112.9 million of horticulture products import; $156.8 million of horticulture products export | (Horticulture Innovation Australia, 2016) |
| Gross Value and Local Value of Agricultural Commodities | SA4 region | Over 60 agricultural commodities | 2007-08 to 2014-15 | $1,119 million gross value of agricultural commodities produced in Mackay in 2014-15 | (Australian Bureau of Statistics, 2013b, 2015), (Australian Bureau of Statistics, 2013a), (Australian Bureau of Statistics, 2016b) |

* GRP - Gross Regional Product; ** SA4 – Statistical Area 4



## 4. Results and Discussion

Tropical Cyclone Debbie affected more than 10,000 jobs, and caused a loss in value added of more than 3 billion AUD. These results are of the same order of magnitude as post-cyclone damage estimates made by the Queensland Treasury (Tapim, 2017), and include indirect impacts, such as employment losses in businesses serving the tourist industry (Reynolds, 2017).

Employment losses are expressed in terms of *full-time equivalent (FTE) employment temporarily affected*. The time span of these losses may range between a number of weeks (for example for coal mines that could be re-opened soon after the cyclone (Ker, 2017; Robins, 2017)) to one year (for example tree crops that will not yield until one year later).

### 4.1 Regional spill-overs

Not surprisingly, Tropical Cyclone Debbie wreaked the most intense havoc where it made landfall, in the regions of Mackay, Fitzroy, and Northern Queensland, and where heavy rains caused widespread flooding, around Brisbane and in Northern New South Wales (the Richmond-Tweed statistical region; Fig. 1, bottom). There is not a single region in the remainder of Australia that is unaffected by the cyclone. In the multi-region input-output disaster model in Eq. (2), these spill-overs come about because businesses experiencing production losses are unable to supply their clients, and also cancel orders for their own inputs, thus leaving businesses elsewhere with reduced activity. In the following sections, we will examine the nature of these spill-overs, by production layer (Section 3.2; see Eq. (4)) and by detailed products and supply-chains (Section 3.3).




Fig. 1: Geographical spread of the value-added (*VA*) loss $\Delta Q_{VA} = \mathbf{q}_{VA}(\mathbf{I} - \mathbf{A}_{\text{ess}})^{-1}\Delta\mathbf{y}$ caused by Tropical Cyclone
Debbie. A comparison of our results (top, $-\log_{10}(VA)$, with *VA* in AU$m) with a satellite image of the cyclone (bottom) shows losses
in northern Queensland regions as a direct consequence of the destructive winds, and losses in southern Queensland and northern NSW
as a result of heavy rain and floods occurring in the cyclone's wake. *Region acronyms:* RemNSW: *Rest of New South Wales (NSW);*
NSW-Rm&T: *NSW Richmond & Tweed;* VIC: *Victoria;* QLD-B: *Queensland (QLD) – Brisbane;* QLD-WBB: *Wide Bay Burnett;*
QLD-DD: *Darling Downs,* QL-SW: *South West;* QLD-F: *Fitzroy;* QLD-CW: *Central West;* QLD-M: *Mackay;* QLD-N: *Northern;*
QLD-FN: *Far North;* QLD-NW: *North West;* SA: *South Australia;* WA: *Western Australia;* TAS: *Tasmania;* ACT: *Australian*
*Capital Territory;* NT: *Northern Territory.*




4.2  Spill-overs and impact sequence
The production layer decomposition defined in Eq. (4) indicates how the impacts of the cyclone unfolded regionally.
Production layer 1 (see section 3.2 for an explanation of production layers) indicates that the tally of value added losses
in directly affected regions was about 2 billion AUD (Fig. 2), which derives straight out of our scenario definition
described in Section 3.3. In addition, the cyclone caused another 1 billion AUD of value added lost across the supply-
chain network of the directly affected businesses. About 7,000 jobs were directly affected, and an additional 5,000
indirectly. The combined sectoral and regional spill-overs are therefore significant: about 50% of value-added lost, and
about 70% of affected employment. Regional spill-overs alone into areas totally unaffected by the cyclone itself are in
the order of 10%.

Whilst the coastal areas of Northern Queensland, Mackay, Fitzroy, Brisbane (in South Queensland) and Northern New
South Wales (Richmond-Tweed area) were affected immediately through storm and flood damage, repercussions were
subsequently felt in the rest of the affected regions, and later on within the rest of Australia. Losses in value added and
employment cascaded throughout inter-regional supply-chains, as subsequent transactions were cancelled. Shortfalls
were noticeable even by distant suppliers, removed from directly affected producers by four or more transaction nodes
(Fig. 2).








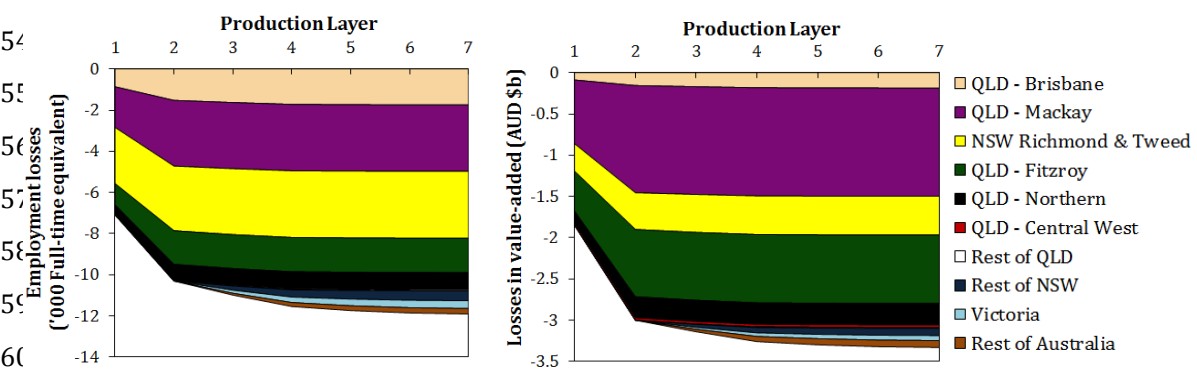


Fig. 2: Sequence of the losses in value added and employment resulting from Tropical Cyclone Debbie for various
regions. The figure shows the first seven production layers, which are upstream layers of the supply chain.





### 4.3 Spill-overs by region and sector

Whilst only a few industries were directly affected by the storm and flooding (coal, tourism, sugar cane, road transport, vegetable growing; black stripes in Fig. 3), many more industries were affected indirectly. Based on a detailed structural path analysis (Defourny and Thorbecke, 1984; Crama et al., 1984), we identified detailed inter-regional repercussions. The shutdown of coal mining in Queensland's Mackay and Fitzroy regions potentially affected more than 100 jobs in finance, property and business services sectors in New South Wales and Victoria; construction industries around Brisbane and in NSW; and business servicing the mining sector as far as Western Australia. Similarly, the destruction of tourist infrastructure in the Richmond-Tweed area affected employment in the food manufacturing sector across the rest of New South Wales.

The top-ranking industries affecting employment directly and elsewhere are those connected to tourism (such as accommodation, restaurants, recreational services, and retail trade, Table 4), Fig 3. In the Richmond-Tweed area of New South Wales, 2500 jobs were affected directly in accommodation, cafes and trade, and about 650 indirectly in other industries and regions due to supply-chain effects (spill-over). Similar effects are observed in Mackay and Brisbane in Queensland. The temporary coal mine shutdown in Mackay and Fitzroy affected even more jobs indirectly than directly. The same holds for the flattening of sugar cane crop in Mackay, and other agriculture (Macadamia and almond nuts) in the Richmond and Tweed rivers that were heavily flooded. Damaged and closed roads affected road transport establishments, and almost equally the industries that depended on them.

### 4.4 Implications for disaster recovery plans

Analysis of the impacts of disasters, such as undertaken in this paper, can have constructive uptake by informing disaster recovery plans as well as regional plans more generally. The Queensland Government management review of Cyclone Debbie in August 2017 recommended improved Business Continuity Planning (BCP) as a way to build : *"... business and organisational resilience [...] Enhanced BCP within state agencies, businesses and communities will help all to be more resilient to the impact of events. [...and..] should feature permanently in disaster management doctrine."* In addition, the report noted that *"BCP needs to consider supply chains, and the numbers and skills of frontline staff required to ensure functioning of critical services"* (IGEM, 2017). Consideration of the large indirect impacts identified in this article, would help improve future Business Continuity Planning. This could be achieved, for example, by considering the large number of employees indirectly affected by the disaster (Table 4), and the related services and products they provide.





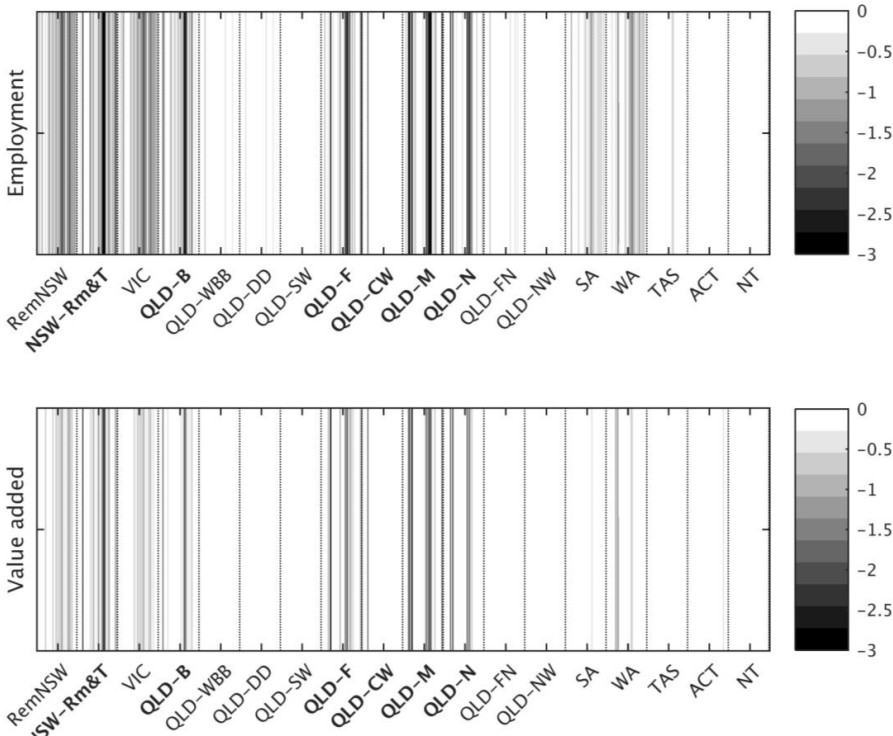


Fig. 3: Employment and value added effects resulting from a tropical cyclone, modelled for threshold $\theta =1.4\times10^{-2}$. The
magnitude of employment and value-added losses is expressed as $\log_{10}|\Delta Q|$ and visualised as lines on a grey scale. Each
line represents one of the 34 industries in each region, in the sequence order listed in Supplementary Information *S1*.
Region acronyms as in Fig. 1, bold regions are those directly affected.









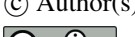



Table 4: Direct, indirect and total employment affected, by causing sector.

| Region | Sector | Direct employment impacts (FTE) | Indirect employment impacts (FTE) | Total employment impacts (FTE) |
|---|---|---|---|---|
| NSW – Richmond-Tweed | Accommodation, cafes and restaurants | -1551 | -278 | -1829 |
| NSW – Richmond-Tweed | Trade | -966 | -371 | -1337 |
| QLD - Brisbane | Trade | -699 | -540 | -1239 |
| QLD - Mackay | Coal, oil and gas | -428 | -712 | -1140 |
| QLD - Fitzroy | Coal, oil and gas | -256 | -498 | -754 |
| QLD - Mackay | Accommodation, cafes and restaurants | -363 | -248 | -611 |
| QLD - Fitzroy | Trade | -395 | -136 | -531 |
| QLD - Mackay | Sugar cane growing | -208 | -244 | -453 |
| QLD - Mackay | Trade | -328 | -110 | -437 |
| QLD - Mackay | Other agriculture | -215 | -143 | -359 |
| QLD - Mackay | Road transport | -200 | -157 | -357 |
| QLD - Northern | Trade | -197 | -97 | -295 |
| QLD - Northern | Sugar cane growing | -106 | -143 | -249 |
| QLD - Fitzroy | Personal and other services | -147 | -94 | -241 |
| QLD - Mackay | Cultural and recreational services | -155 | -84 | -239 |
| QLD - Brisbane | Accommodation, cafes and restaurants | -119 | -114 | -233 |
| NSW - Richmond & Tweed | Other agriculture | -92 | -139 | -231 |
| QLD - Fitzroy | Residential building construction | -89 | -136 | -225 |
| QLD - Fitzroy | Road transport | -100 | -93 | -193 |
| QLD - Mackay | Residential building construction | -77 | -110 | -187 |
| **Total** | | -7092 | -4878 | -11970 |


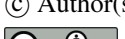



Table 5: Direct, indirect and total value added affected, by causing sector.

| Region | Sector | Direct value added impacts (AUD $m) | Indirect value added impacts (AUD $m) | Total value added impacts (AUD $m) |
|---|---|---|---|---|
| QLD - Mackay | Coal, oil and gas | -519 | -472 | -990 |
| QLD - Fitzroy | Coal, oil and gas | -360 | -332 | -692 |
| NSW - Richmond & Tweed | Accommodation, cafes and restaurants | -148 | -69 | -217 |
| NSW - Richmond & Tweed | Trade | -107 | -72 | -178 |
| QLD - Northern | Coal, oil and gas | -71 | -70 | -141 |
| QLD - Brisbane | Trade | -64 | -53 | -117 |
| NSW - Richmond & Tweed | Other agriculture | -64 | -45 | -110 |
| QLD - Mackay | Sugar cane growing | -41 | -47 | -88 |
| QLD - Mackay | Other agriculture | -48 | -29 | -78 |
| QLD - Mackay | Accommodation, cafes and restaurants | -40 | -28 | -68 |
| QLD - Fitzroy | Trade | -40 | -23 | -63 |
| QLD - Mackay | Cultural and recreational services | -42 | -20 | -62 |
| QLD - Northern | Sugar cane growing | -25 | -31 | -56 |
| QLD - Mackay | Trade | -35 | -14 | -49 |
| QLD - Fitzroy | Residential building construction | -21 | -26 | -46 |
| QLD - Fitzroy | Personal and other services | -27 | -17 | -43 |
| QLD - Northern | Trade | -27 | -14 | -41 |
| QLD - Mackay | Road transport | -20 | -20 | -40 |
| QLD - Mackay | Residential building construction | -20 | -20 | -40 |
| QLD - Northern | Residential building construction | -19 | -18 | -37 |
| **Total** | | **-1849** | **-1493** | **-3342** |




**5.    Conclusions and outlook**

Powerful tropical cyclones have the ability to cause severe disruptions of economic production that are felt far beyond
the areas of landfall and flooding. Here, we used an input-output-based analytical tool for enumerating the post-disaster
consumption possibilities, and ensuing direct and indirect losses of employment and value added as a consequence of the
Tropical Cyclone Debbie that hit the Queensland regions of Australia in March and April 2017. Our work contributes an
innovative approach for (a) quantifying the impact of disasters in a detailed and timely manner and (b) incorporating
infrastructure damages into the assessment of losses in employment and value-added. Our approach can be applied to
other regions, and ultimately extended to include impacts well beyond employment and value added, such as wider
environmental or social consequences of disasters. The IELab already has many satellite accounts (and is being
expanded) to assess broader environemntal and social flow-on effects. The growing number of Virtual Laboratories for
input-output analysis (Geschke and Hadjikakou, 2017) for countries in disaster-prone zones (Indonesia, Taiwan, China)
means that the work described in this paper is directly transferrable to other geographical settings.

Our results from this Australian case study suggest that Cyclone Debbie caused substantial damage to spill over into
regions and sectors not directly affected: Industries directly hit by the cyclone suffered approximately 7000 job losses,
but some 5000 jobs (another 70%) were affected in these industries' supply chains. A total of $3 billion losses in value
added were observed, $2 billion of which were direct with particular impact around Mackay and Fitzroy, as well as the
coastal areas of Northern Queensland, Brisbane and northern New South Wales (Richmond-Tweed area). These findings
demonstrate that the full supply-chain effects of major disruptions on national economies are significant, and that this
type of study will become increasingly important in a future likely to be fraught with extreme weather events, as the
frequency and intensity of tropical cyclones increase as a result of climate change (Mendelsohn et al., 2012).

This work demonstrates rapid analysis of the wide indirect impacts of Cyclone Debbie. It shows how significant
consequences can be felt, as spill-overs, in regions well outside the landfall and flood zones caused by the cyclone. Our
work suggests improved planning could help account for these impacts, minimise them in future, and thereby help
transition the affected economies towards greater resilience. For example, about 1200 employees providing services to
coal mines were affected by Cyclone Debbie, however this impact is currently not mentioned in the disaster recovery



planning. Greater consideration of the influence of mine shutdown on supply chains could be an important future element
of Business Continuity Planning.

*5.1 Outlook*
In this work, we have examined losses of employment and value added, because these are currently of immediate
importance for governments, insurers and the media. Future work could investigate possibilities for re-structuring the
geography of production and supply-chain networks with the aim of finding more "disaster-resilient" configurations. In
addition, there are variants of input-output-analytical methods that allow establishing optimal recovery paths (Koks et al.,
2016), and these approaches could be integrated into the Australian Industrial Ecology Virtual Laboratory.

Future work could also consider the effects of cyclones beyond national borders. The disruptions of coal exports due to
Tropical Cyclone Debbie, for example, caused bottlenecks in Indian and Chinese steel mills (The Barrel, 2017), and
during the aftermath of the storm, steel producers were looking for alternative sources of coal such as Russia, Mongolia
or Mozambique (Serapio, 2017). Such trade relationships can be taken into account using nested, multi-scale, global
multi-region input-output frameworks (Bachmann et al., 2015; Wang et al., 2015; Tukker and Dietzenbacher, 2013).







**Competing interests**

The authors declare that they have no conflict of interest.





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
