# Peer review of "Economic damage and spill-overs from a tropical cyclone"

_Natural Hazards and Earth System Sciences, 2017_

## Referee Comment (RC1) · Anonymous Referee #1 · 25 Jan 2018

**General comments**

In general the writing is okay, but there are few logical gaps in some paragraphs. Some terms are not described and explained in sufficient detail. The manuscript is not very well structured, which leads to redundancy in some parts and makes it hard to follow in other parts. From my point of view the adopted methodology to estimate the indirect damage is sound. However the description and explanations of the methods should be more extensive. Graphics and tables are only partly clear and not all of them support the understanding of main contents. Results are innovative as, to my knowledge, there are no studies dealing with indirect consequences of tropical cyclone per sector on a regional scale. Also the authors attempt to include the infrastructure damage in the estimation of the indirect consequences. However, the presentation of the results

could be improved. In addition the results are not discussed sufficiently. Especially the reliability of the estimates of the direct damage should be addressed. The conclusions of the manuscript are only partly supported by the results. In general, I think the topic is appropriate for publication in NHESS, but the manuscript needs some major revisions.

**Specific comments**

Introduction

- From my point of view the introduction could be improved by a general reorganization. At the moment two thirds of the introduction is a review/summary of the event based on newspapers and government papers followed by a paragraph describing the methods used in the study without giving an introduction to the methods. I suggest to combine section 1 and 2. With a clearly defined review of the event and a review of input-output analysis.

- Please check how media should be cited. In the current version it is hard to check which references are cited, since the names cited in the text are not listed in the reference list.

- Pg. 3 line 65-66: Please clarify that the cyclone names are no citations.

- Pg. 4 line 89: It is not clear at this point what spill-overs are. Please clarify.

- Pg. 5 line 146-149: Please be a bit more specific about the novelty of your research. From my point of view the novelty is not the applied method (which was applied before in e.g. Schulte in den Bäumen 2015, as you mention as well) but rather the case-study as this method was not applied to cyclones before.

Methods

- Pg. 6 line 156-164: This paragraph could rather be shifted to the introduction. I suggest to give a short overview of section 3 instead.
- From my point of view an explanation of the basic input-output equation and a clear naming of the symbols used in the formulas would help readers who are not so familiar with the methodology (e.g. $\mathbf{A}$ is named "matrix of input coefficients" in line 183 and "production recipe" in line 244).

- Pg. 6 eq. 1: Please give more information about the calculation of the Gamma-matrix.

- Pg. 6 eq. 2: For me this equation is not quite clear. Please elaborate a bit further why you max the sum of the vector of the post-disaster consumption possibilities $\mathbf{y_1}$ . Also I could not find information about this equation in Steenge and Bockarjova (2007).

- Pg. 7 line 193: I could not find further details to the suggested input-output approach in Supplementary Information S2. Please give more information about the approach.

- Pg. 8 line 229-230: "the reduction of total industry output (in 2017 compared to 2016)" How is the total industry output 2017 calculated?

Data

- Pg. 15 Fig. 1: Please include unit labels in map and the source of the satellite image

- Please explain what superior means in this context. What makes superior economic data superior?

Results and Discussion

- From my point of view this Section does not seem to be very well organized. In addition not all of the figures and tables are helpful and supporting. A discussion

of the results, e.g. a comparison with results from other studies or a discussion of the reliability of the direct damage estimations is missing entirely. Also it should be more emphasized that the presented results are results produced by a model (at least the indirect damage) and might not necessarily fit to the damage which really occurred in the regions.

- Pg. 14 line 3-10: From my point of view these two paragraphs do not belong here. I suggest a short overview of Section 4 at this point.

- Pg. 14 line 3: Please indicate if these numbers are results from your models.

- Pg. 14 line 8: Please explain the term "full-time equivalent"

- Pg. 14 line 20-21: "…detailed products and supply-chains (Section 3.3)." Please be more specific about this reference.

- Pg. 17 line 70-71: "…detailed structural path analysis…" Please be more specific about this analysis. How did you apply it? What does the analysis exactly do? Etc. Also this should be mentioned in Section 3.

- Fig. 2: Please choose a different color for "Rest of QLD" than white. At a first look it is a bit confusing with the free space at the bottom of the plots.

- Fig. 3: Please improve this figure. Currently it is very hard to get information about the damage to different industries. Hence at least for the value added this is basically the same information as shown in Fig. 1. Also it is not clear what the threshold theta means? Please clarify.

- Table 4 and 5: How do you differentiate between the direct and indirect impacts? Since the direct effects are included in the estimation of the indirect effects

- Table 5 is not mentioned in the text.

[Figure]

Conclusions and outlook

- Pg. 21 line 148-150: "...Our approach can be applied to other regions, and ultimately extended to include impacts well beyond employment and value added, such as wider environmental or social consequences of disasters ..." From my point of view this is not sufficiently supported by the results.

- Pg.21 line 167-169: "...For example, about 1200 employees providing services to coal mines were affected by Cyclone Debbie, however this impact is currently not mentioned in the disaster recovery planning. ..." This is hardly shown in the results.

**Technical corrections**

- Please be consistent with the abbreviations you introduce. E.g. in line 246 you write "...input-output (IO) databases." although IO was already introduced.

- Please mention and explain all figures and tables included in the manuscript also in text

- Pg. 11 line 245: "... (IELab; (Lenzen et al., 2014)." A bracket is missing

- Pg. 11 line 249: "different regions (Tukker and Dietzenbacher, 2013); see (Leontief, 1953) for an account of MRIO theory)." A bracket is missing

- Pg. 21 line 151: "...broader environemntal ..." please correct typo.

- Pg. 22 line 176: There are two references for Koks et al. 2016. Please indicate which one is cited here.

---

## Referee Comment (RC2) · Anonymous Referee #2 · 8 Feb 2018

This is a straightforward paper with a clear structure and presentation. The highlights of this paper appear "(a) quantifying the impact of disasters in a detailed and timely manner and (b) incorporating infrastructure damages into the assessment of losses in employment and value-added", as written in the conclusion section. As for (a), it may be the first model/paper utilizing the multiregional Australian input-output table with 19 regions and 34 industries, while the process for producing such detailed input-output tables were described in other papers (page 11). So, what's new in this regard seems to be the use of the superior economic data (in sub-section 3.4.1) written in one paragraph and table 3. It seems to me if this is one of the main contributions of the paper, it should be discussed more thoroughly, if such contents are available.

In terms of (b), it is described in sub-section 3.3.2, in which they indicated that their

method for this is similar to Hallegatte (2008), as written in page 8. There have been more sophisticated and/or complicated modeling frameworks to incorporate infrastructure damages with input-output analysis for disaster impact analysis, such as Tsuchiya et al. (2007) referred in this paper. So, again, this is not completely new here, either.

Moreover, their detailed multiregional input-output table is used in the rather standard way, as described in pages 6-7, with the Steenge and Bockarjova (2007) approach. There seems no new trick here, either. At the same time, the issues of input-output analysis for disaster impact analysis have been discussed and were summarized well in Oosterhaven (2017), in which he claimed six aspects of disaster impact and argued that input-output analysis covers only a subset of those six aspects. Since this paper also use the standard input-output analysis, the results of this paper should cover only the limited extent of the disaster impacts. At least, this should be discussed, and hopefully would be incorporated in the revised version.

Furthermore, since this paper focuses on the changes in consumption and value-added, the Miyazawa's enlarged input-output framework should be also discussed and would be included for the comparison of the results.

Oosterhaven, J. (2017) On the limited usability of the inoperability IO model. Economic Systems Research, 29: 452-461.

---

## Author Comment (AC1) · 14 Apr 2018

The comment was uploaded in the form of a supplement:
https://www.nat-hazards-earth-syst-sci-discuss.net/nhess-2017-440/nhess-2017-440-AC1-supplement.pdf
* * *

---

## Author Comment (AC2) · 14 Apr 2018

**Reviewer #2**

**Comment:** This is a straightforward paper with a clear structure and presentation. The highlights of this paper appear "(a) quantifying the impact of disasters in a detailed and timely manner and (b) incorporating infrastructure damages into the assessment of losses in employment and value-added", as written in the conclusion section. As for (a), it may be the first model/paper utilizing the multiregional Australian input-output table with 19 regions and 34 industries, while the process for producing such detailed input-output tables were described in other papers (page 11). So, what's new in this regard seems to be the use of the superior economic data (in sub-section 3.4.1) written in one paragraph and table 3. It seems to me if this is one of the main contributions of the paper, it should be discussed more thoroughly, if such contents are available.

**Response:** Regarding primary (superior) economic data we have added information in Section 2.5.1. including wider referencing of key information sources eg (Queensland Treasury and Trade, 2013, and Wilkinson, J., 2014) which contain full details of data used as constraints. We have not included these data as supplementary information as we did not want to double-up with the original authors' work, however key information could be appended if that is necessary.

**Comment:** In terms of (b), it is described in sub-section 3.3.2, in which they indicated that their method for this is similar to Hallegatte (2008), as written in page 8. There have been more sophisticated and/or complicated modeling frameworks to incorporate infrastructure damages with input-output analysis for disaster impact analysis, such as Tsuchiya et al. (2007) referred in this paper. So, again, this is not completely new here, either.

Moreover, their detailed multiregional input-output table is used in the rather standard way, as described in pages 6-7, with the Steenge and Bockarjova (2007) approach. There seems no new trick here, either. At the same time, the issues of input-output analysis for disaster impact analysis have been discussed and were summarized well in Oosterhaven (2017), in which he claimed six aspects of disaster impact and argued that input-output analysis covers only a subset of those six aspects. Since this paper also use the standard input-output analysis, the results of this paper should cover only the limited extent of the disaster impacts. At least, this should be discussed, and hopefully would be incorporated in the revised version.

Furthermore, since this paper focuses on the changes in consumption and value-added, the Miyazawa's enlarged input-output framework should be also discussed and would be included for the comparison of the results.

Oosterhaven, J. (2017) On the limited usability of the inoperability IO model. Economic Systems Research, 29: 452-461.

**Response:** We have cited the references that the reviewer lists, and we have added additional clarification to section 2.4.2, broadening our referencing to the work of others, and helping to clarify the key contribution of this work. In particular, we draw attention to recent acknowledgement of several authors that the preferred method for inclusion of infrastructure in disaster impact analysis is a continuing question.

In response to the comments, we have added the following text in the manuscript:

*In compiling the gamma matrices, damages were only considered where we could find empirical monetary information. With respect to modelling the effect of capital infrastructure damages on production, we were bound by the gamma-matrix formalism of the Steenge-Bočkarjova method. We note that other more detailed and sophisticated modelling frameworks have been used, such as Tsuchiya et al. (2007).*

*Finally, beneficial effects can result from natural disasters. In Queensland for example, the replacement or repairs to damaged buildings and infrastructure, or any other demand for commodities required especially for post-disaster recovery, is likely to have created additional employment and value added and may have spawned technology updates. In addition, above-average rainfall may have been beneficial for pastures and water supply, and increased freshwater run-off and turbidity could have increased catches of prawn trawling. As no data were available for quantifying such repercussions, these effects are not accounted for in our study.*

*Steenge and Bočkarjova (2007) remarks that a preferred method for disaster impact analysis does currently not exist, due to (a) many possible research questions, and (b) many relevant items of information surrounding disasters being unknown. Steenge and Bočkarjova (2007) also clarify the strengths and weaknesses of static input-output analysis against dynamic CGE modelling. In this context, they warn against overly optimistic assumptions regarding market flexibility and substitution. Oosterhaven (2017) summarises the shortcomings of input-output-based disaster analysis approaches in their attempt to estimate real-world consequences of disasters.*

---

## Author Response (AR1)

**Reviewer #1**

**General comment:** In general the writing is okay, but there are few logical gaps in some paragraphs. Some terms are not described and explained in sufficient detail. The manuscript is not very well structured, which leads to redundancy in some parts and makes it hard to follow in other parts. From my point of view the adopted methodology to estimate the indirect damage is sound. However the description and explanations of the methods should be more extensive. Graphics and tables are only partly clear and not all of them support the understanding of main contents. Results are innovative as, to my knowledge, there are no studies dealing with indirect consequences of tropical cyclone per sector on a regional scale. Also the authors attempt to include the infrastructure damage in the estimation of the indirect consequences. However, the presentation of the results could be improved. In addition the results are not discussed sufficiently. Especially the reliability of the estimates of the direct damage should be addressed. The conclusions of the manuscript are only partly supported by the results. In general, I think the topic is appropriate for publication in NHESS, but the manuscript needs some major revisions.

**General Response:** We thank the reviewer for their comments. We have now revised the manuscript, as per the specific comments below. Furthermore, we provide updated figures and tables in the revised manuscript. Regarding "reliability of estimates of direct damage", we note that in section 2.4.1, and accompanying Supplementary Information 2.2 and 2.3 we acknowledge the wide range of sources we have drawn on to characterise the direct damage in both monetary and non-monetary terms. We accept the point made by reviewer 2 and have therefore added *"Nevertheless, the numbers should be treated as estimates, and ideally validated when the actual final costs are known. However, such detailed validation of disaster impacts is problematic with the rapid impact assessment which is the focus of this research."*

**Specific comments:**
- From my point of view the introduction could be improved by a general reorganization. At the moment two thirds of the introduction is a review/summary of the event based on newspapers and government papers followed by a paragraph describing the methods used in the study without giving an introduction to the methods.

**Response:** *Introduction restructure*: Currently the introduction (926 words) is structured as follows:
- 165 words event description, followed up with
- 128 words physical damage description, followed up by
- 89 words reinforcing that TC Debbie is not an isolated event
- 342 words describing significance of economic damage, followed up with
- 71 words reinforcing the importance of tools for assessing regional damage comprehensively and for future planning
- 280 words describing our aim and approach, wrapping up with
-  85 words setting the scene for the paper.

Hence, only 1/3 of the introduction describes the actual event, whereas 89+342+71 words are used for arguing that cyclones and associated floods is and will remain an important problem for Australia as a whole. We think that this information needs to be included in the introduction.

Nevertheless, we have now
- inserted some key introduction phrases to paragraphs to clarify their purpose within the introduction.

- moved the fifth paragraph into the Methods section, where – as the referee pointed out – the matching details are described.
- shifted a methods paragraph that the referee identified into the introduction as requested.

- I suggest to combine section 1 and 2. With a clearly defined review of the event and a review of input-output analysis.

**Response:** Conventional journal style distinguishes a non-technical introduction and a technical methods section. Given that the previous Section 2 was rather detailed, we have therefore combined Section 2 with Section 3 to form a new Section 2: Methods. This new methods section is now preceded with a short paragraph explaining to the reader what is to be expected.

- Please check how media should be cited. In the current version it is hard to check which references are cited, since the names cited in the text are not listed in the reference list.

**Response:** We follow the Endnote citation style of Natural Hazards and Earth Systems Sciences. We have removed the footnotes and have included references for media in the main reference list.

- Pg. 3 line 65-66: Please clarify that the cyclone names are no citations.

**Response:** We have edited the text as follows *"Severe tropical cyclones such as Debbie are not an isolated phenomenon. Past tropical cyclones, in Australia and elsewhere, have disrupted food supply – for example in Madagascar (Cyclone Gafilo in 2004), Vanuatu (Cyclone Pam in 2015), and Fiji (Cyclone Winston in 2016)."*

- Pg. 4 line 89: It is not clear at this point what spill-overs are. Please clarify.

**Response:** We have now clarified what spill-overs are. We write: *"First, we are able to examine identify the consequences of the cyclone not only for the directly affected regions and industry sectors, but for the wider Australian economy. Such indirect effects are caused by damaged businesses ceasing to supply clients, or ceasing to require inputs from suppliers. Thus, economic activity winds down elsewhere as well. Such effects are called (regional and sectoral) spill-overs"*.

- Pg. 5 line 146-149: Please be a bit more specific about the novelty of your research. From my point of view the novelty is not the applied method (which was applied before in e.g. Schulte in den Bäumen 2015, as you mention as well) but rather the case-study as this method was not applied to cyclones before.

**Response:** We have edited the text as follows: *"One particular type of disaster IO analysis, proposed by Steenge and Bočkarjova (2007) aims at investigating post-disaster consumption possibilities as a consequence of production shortfalls resulting from a disaster. Such an assessment has been applied, for example to widespread flooding in Germany (Schulte in den Bäumen et al., 2015) and electricity blackouts from possible severe space weather events (Schulte in den Bäumen et al., 2014). Here, we apply this method for the first time to undertake an estimation of post-disaster consumption possibilities, and subsequent losses in employment and economic value added resulting from the 2017 Tropical Cyclone Debbie in Australia."*

- Pg. 6 line 156-164: This paragraph could rather be shifted to the introduction. I suggest to give a short overview of section 3 instead.

**Response:** Done. Thank you.

- From my point of view an explanation of the basic input-output equation and a clear naming of the symbols used in the formulas would help readers who are not so familiar with the methodology (e.g. A is named "matrix of input coefficients" in line 183 and "production recipe" in line 244).

**Response:** We write that "*Constraint i) in Equation (2) is the standard fundamental input-output accounting relationship stating that in every economy intermediate demand $T$ and final demand $y$ sum up to total output $x$. This can be seen by writing $y_1 = (I - A)x_1 = x_1 - T1 \Leftrightarrow T1 + y_1 = x_1$* ." We have also changed "production recipe" to "input coefficients matrix" to be consistent in our naming of variables.

- Pg. 6 eq. 1: Please give more information about the calculation of the Gamma-matrix.

**Response:** We now write "The entries of $\Gamma$ are populated on the basis of primary data, in our case about cyclone Debbie (Section 2.4).".

- Pg. 6 eq. 2: For me this equation is not quite clear. Please elaborate a bit further why you max the sum of the vector of the post-disaster consumption possibil-ities y1 . Also I could not find information about this equation in Steenge and Bockarjova (2007).

**Response:** The problem in Steenge and Bockarjova is that final demand can become negative. Consider their pre-disaster equation 21: Change Gamma to $\begin{matrix} 0.2 & 0 \\ 0 & 0.8 \end{matrix}$ and you obtain

| 0.25 | 0.4 | x | 20 | + | -1 | = | 20 |
|------|-----|---|----|---|------|---|----|
| 0.14 | 0.12 | | 40 | | 32.4 | | 40 |

Maximising y and at the same time ensuring non-negativity yields

| 0.25 | 0.4 | x | 20 | + | 0 | = | 20 |
|------|-----|---|------|---|------|---|------|
| 0.14 | 0.12 | | 37.5 | | 30.2 | | 37.5 |

We have now described this in the main text.

- Pg. 7 line 193: I could not find further details to the suggested input-output approach in Supplementary Information S2. Please give more information about the approach.

**Response:** This is simply an error. S2 should be references when describing the population of the Gamma matrix. This is now corrected.

- Pg. 8 line 229-230: "the reduction of total industry output (in 2017 compared to 2016)" How is the total industry output 2017 calculated?

**Response:** For this we have clarified by assuming 2017 output equal to 2016 output minus output losses attributable to the cyclone.

- Pg. 15 Fig. 1: Please include unit labels in map and the source of the satellite image

**Response:** We have included the units for the numbers (-2.5, -2 etc.) in the legend. We have now included the source of the satellite image.

- Please explain what superior means in this context. What makes superior eco-nomic data superior?

**Response:** "Superior" is a term used amongst statisticians and IO table compilers, for primary data used in reconciliation exercises where most of the information is estimated. Here, data taken from external sources is relatively superior to estimated table entries. We recognise that this detail is lost here, so we have replaced "superior" by "primary", to indicate that the data are from the primary source, or "raw".

- From my point of view this Section does not seem to be very well organized. In addition not all of the figures and tables are helpful and supporting. A discussion of the results, e.g. a comparison with results from other studies or a discussion of the reliability of the direct damage estimations is missing entirely.

**Response:** We have restructured this Results and Discussion section and substantially revised to clarify the content and interpretation of the primary results. This includes giving an overview of this section at the beginning, presenting on the overall results first, following by more detailed analysis of spill-overs, and discussing the implications for disaster recovery plans. In the text, we have also referring to the figures and tables more to better direct the readers to the corresponding figures and tables. In addition, a brief discussion on the reliability of the direct damage estimations is added in the Outlook section. The information on the direct damage estimations used in this study was based on the best available estimations from the government and industry agencies.

- Also it should be more emphasized that the presented results are results produced by a model (at least the indirect damage) and might not necessarily fit to the damage which really occurred in the regions.

**Response:** We have added in Section 3.1 "As one referee noted, our results for indirect damage are produced by a model and might not necessarily reflect damage that really occurred in the regions. However, an application of the same model to a case study where indirect effects were known (see Fig. 5 in Lenzen *et al.* 2017a) shows that measured outcomes were reproduced with reasonable accuracy."

- Pg. 14 line 3-10: From my point of view these two paragraphs do not belong here. I suggest a short overview of Section 4 at this point.

**Response:** We have now moved these two paragraphs into section 3.2, which presents the major results. A short overview of this Results and Discussion Section has been added.

- Pg. 14 line 3: Please indicate if these numbers are results from your models.

**Response:** We now write *"Our results show that"*

- Pg. 14 line 8: Please explain the term "full-time equivalent"

**Response:** We have included the following sentence *"Full-time equivalent means that part-time jobs are expressed as fractional full-time jobs, so that they added into a total."*

- Pg. 14 line 20-21: ". . .detailed products and supply-chains (Section 3.3)." Please be more specific about this reference.

**Response:** This was an error, we have now removed this reference.

- Pg. 17 line 70-71: ". . .detailed structural path analysis. . ." Please be more specific about this analysis. How did you apply it? What does the analysis exactly do? Etc. Also this should be mentioned in Section 3.

**Response:** We have now removed this reference and expanded on the section on production layer decomposition.

- Fig. 2: Please choose a different color for "Rest of QLD" than white. At a first look it is a bit confusing with the free space at the bottom of the plots.

**Response:** We have now used the colour pink for "Rest of QLD".

- Fig. 3: Please improve this figure. Currently it is very hard to get information about the damage to different industries. Hence at least for the value added this is basically the same information as shown in Fig. 1.

**Response:** We have used a different colour scheme to improve this figure. Instead of black and white, we now use the colour scheme 'bone' in MATLAB to highlight the bands. The underlying data for this figure is in the supplementary information section.

- Also it is not clear what the threshold theta means? Please clarify.

**Response:** We have deleted this as this was a left-over from the Appendix where this is listed in Section SI2.

- Table 4 and 5: How do you differentiate between the direct and indirect impacts?Since the direct effects are included in the estimation of the indirect effects

**Response:** The direct effects are $\mathbf{q}\Delta\mathbf{y}$ (Equation 4), and the indirect effects include all other supply chain layers. The values for indirect effects do not include the values for direct effects. Direct and indirect effects added together give total effects (last column in Table 4 and 5).

- Table 5 is not mentioned in the text.

**Response:** We have now mentioned Table 5 in the text.

- Pg. 21 line 148-150: ": : :Our approach can be applied to other regions, and ultimately extended to include impacts well beyond employment and value added, such as wider environmental or social consequences of disasters : : :" From my point of view this is not sufficiently supported by the results.

**Response:** We have moved this statement and the "Outlook" Section from the Conclusion section to the Discussion section 3.5 for a general discussion on possible future work.

- Pg.21 line 167-169: ": : :For example, about 1200 employees providing services to coal mines were affected by Cyclone Debbie, however this impact is currently not mentioned in the disaster recovery planning. : : :" This is hardly shown in the results.

**Response:** We have added description in section 3.4 to direct readers to Table 4 for the results. "*For instance, as shown in Table 4 for the indirect employment impacts for the "Accommodation, cafes and restaurants" sector, some 1,381 employees providing services were affected. However, this impact is currently not mentioned in disaster recovery planning documents*."

- Please be consistent with the abbreviations you introduce. E.g. in line 246 you write ". . .input-output (IO) databases." although IO was already introduced.

**Response:** We have now fixed this.

- Please mention and explain all figures and tables included in the manuscript also in text

**Response:** All figures and tables are now explained in the text.

- Pg. 11 line 245: ". . . (IELab; (Lenzen et al., 2014)." A bracket is missing

**Response:** We have now fixed this.

- Pg. 11 line 249: "different regions (Tukker and Dietzenbacher, 2013); see (Leon-tief, 1953) for an account of MRIO theory)." A bracket is missing

**Response:** We have now fixed this.

- Pg. 21 line 151: ". . .broader environemntal . . ." please correct typo.

**Response:** We have now fixed this.

- Pg. 22 line 176: There are two references for Koks et al. 2016. Please indicate which one is cited here.

**Response:** It is *Koks, E., Carrera, L., Jonkeren, O., Aerts, J. C. J. H., Husby, T. G., Thissen, M., Standardi, G., and Mysiak, J.: Regional disaster impact analysis: comparing input–output and computable general equilibrium models, Natural Hazards and Earth System Science, 16, 1911-1924, 2016.*

**Reviewer #2**

**Comment:** This is a straightforward paper with a clear structure and presentation. The highlights of this paper appear "(a) quantifying the impact of disasters in a detailed and timely manner and (b) incorporating infrastructure damages into the assessment of losses in employment and value-added", as written in the conclusion section. As for (a), it may be the first model/paper utilizing the multiregional Australian input-output table with 19 regions and 34 industries, while the process for producing such detailed input-output tables were described in other papers (page 11). So, what's new in this regard seems to be the use of the superior economic data (in sub-section 3.4.1) written in one paragraph and table 3. It seems to me if this is one of the main contributions of the paper, it should be discussed more thoroughly, if such contents are available.

**Response:** Regarding primary (superior) economic data we have added information in Section 2.5.1. including wider referencing of key information sources eg (Queensland Treasury and Trade, 2013, and Wilkinson, J., 2014) which contain full details of data used as constraints. We have not included these data as supplementary information as we did not want to double-up with the original authors' work, however key information could be appended if that is necessary.

**Comment:** In terms of (b), it is described in sub-section 3.3.2, in which they indicated that their method for this is similar to Hallegatte (2008), as written in page 8. There have been more sophisticated and/or complicated modeling frameworks to incorporate infrastructure damages with input-output analysis for disaster impact analysis, such as Tsuchiya et al. (2007) referred in this paper. So, again, this is not completely new here, either.

Moreover, their detailed multiregional input-output table is used in the rather standard way, as described in pages 6-7, with the Steenge and Bockarjova (2007) approach. There seems no new trick here, either. At the same time, the issues of input-output analysis for disaster impact analysis have been discussed and were summarized well in Oosterhaven (2017), in which he claimed six aspects of disaster impact and argued that input-output analysis covers only a subset of those six aspects. Since this paper also use the standard input-output analysis, the results of this paper should cover only the limited extent of the disaster impacts. At least, this should be discussed, and hopefully would be incorporated in the revised version.

Furthermore, since this paper focuses on the changes in consumption and value-added, the Miyazawa's enlarged input-output framework should be also discussed and would be included for the comparison of the results.

Oosterhaven, J. (2017) On the limited usability of the inoperability IO model. Economic Systems Research, 29: 452-461.

**Response:** We have cited the references that the reviewer lists, and we have added additional clarification to section 2.4.2, broadening our referencing to the work of others, and helping to clarify the key contribution of this work. In particular, we draw attention to recent acknowledgement of several authors that the preferred method for inclusion of infrastructure in disaster impact analysis is a continuing question.

In response to the comments, we have added the following text in the manuscript:

[revised manuscript text omitted]

---

## Author Response (AR2)

**Response to reviewers**

**We thank the reviewers for their insightful comments, which have strengthened this manuscript.**

**Reviewer:** After reading the authors' responses, I feel that the authors are not particularly interested in disaster impact analysis. Rather, it seems that they just want to show what they can do with their data and model.

I raised the question about the limitations of input-output analysis on disaster impact analysis and referred Oosterhaven (2017) in my previous review comments. They added this reference in the revised version, but appear not reading it carefully. Their method (or Steenge and Bockarjova, 2007) is the standard Leontief model, which is a demand-driven model. They transformed supply shortages (or changes) to demand changes based on equation (2) on page 6, but it was plugged into the standard Leontief model (or somewhat modified with q), which is still a demand driven model. Demand driven model can derive only backward linkage effects, not forward linkage effects, which is one of the claims of Oosterhaven (2017).

If they read Oosterhaven (2017) carefully, they should find Oosterhaven's six classifications of disaster impact: (1) supply shortage due to damages on production facilities including infrastructure, which 'will have forward or downstream effects (page 453)'; 2) supply shock of non-replaceable intermediate inputs; 3) substitution effects on replaceable intermediate inputs; 4) impacts from demand decline due to damages on production facilities including infrastructure; 5) impacts from redistribution of consumption demand; and 6) impacts of reconstruction demand injection. As described in the methodology part of their paper (pages 6-7), it is obvious that this paper's analysis captures only 4) and a part of 5), none of the supply side impacts of 1), 2), and 3), which cannot be derived with the standard Leontief model as made clear in, again, Oosterhaven (2017).

The authors' response argues that because Steenge and Bockarjova (2007), by which the paper's analysis is based on, indicated that a preferred method for disaster impact analysis does not exist, their method in this paper can be useful. Steenge and Bockarjova (2007) paper was published in 2007. Since then, we have come a long way to evaluate and improve disaster impact models to this date, as some of the referenced papers in this paper discuss. The authors review some of the improved models, but they still use the standard Leontief model without carefully warning readers about their severe limitations. I have to insist that not indicating what the model in this paper can derive AND cannot cover could be seen as dishonesty of their attitude toward readers. At least, they need clearly write that this paper's results capture only a small subset of disaster impacts as above, because of the use of input-output model.

**Response:** Thank you for your comments. In light of these comments, we have made the following modifications to the text:

Addition: Lines 160-161: As this method uses Leontief's demand-driven model, it captures backward, upstream supply-chain imapcts resulktimng from a disaster.

Addition: Lines 278-287: First, since this study uses Leontief's demand-driven IOA version, we are only able to quantify backward, or upstream supply-chain effects, such as impacts from decline of demands due to damages to production facilities and changed consumption possibilities. We are unable to quantify the forward or downstream effects of supply-side shocks due to the unavailability of non-replaceable production inputs, or substitution effects due to the unavailability of replaceable production inputs. As such, this study covers only a subset of Oosterhaven (2017) classifications of potential disaster impacts. A more comprehensive, but also significantly more data-hungry approach would be to use dynamic CGE modelling, however in this context Steenge and Bočkarjova (2007) warn against overly optimistic assumptions regarding market flexibility and substitution. A promising way forward is the linear programming approach by Oosterhaven and Bouwmeester (2016) in which the authors minimise the information gain between pre- and post-disaster inter-regional IO tables.

Deletion: Lines 301 – 306: Steenge and Bočkarjova (2007) remark that a preferred method for disaster impact analysis does currently not exist, due to (a) many possible research questions, and (b) many relevant items of information surrounding disasters being unknown. Steenge and Bočkarjova (2007) also clarify the strengths and weaknesses of static input-output analysis against dynamic CGE modelling. In this context, they warn against overly optimistic assumptions regarding market flexibility and substitution. Oosterhaven (2017) summarises the shortcomings of input-output based disaster analysis approaches in their attempt to estimate real world consequences of disasters.

---

## Author Response (AR3)

**Response to reviewers**

**We thank the reviewers for their insightful comments, which have strengthened this manuscript.**

**Reviewer:** The authors responded to my second comments straightforwardly. Now, it is clear what this paper is about. It seems what's new about this paper comparing with the related literature is the use of detailed data set applied to Australian cyclone case. It is just a simple case study.

At the same time, still in the abstract and the introduction section of this paper, a focus of the analysis is to investigate the ripple effect "through entire supply-chain networks" (in abstract). Entire supply-chain network includes its upstream and downstream. As the revised paper indicates, the methodology of this paper can deal only with upstream supply-chain (backward linkage), but not downstream supply-chain. This is just a half of the 'entire supply-chain'. Moreover, as the 2011 East Japan Earthquake and Tsunami case highlighted, impacts via downstream supply-chains could spread over wider areas and across various industrial sectors, especially if infrastructure were damaged. Again, these downstream (forward linkage) impacts are not dealt in this paper, thus it is far from the analysis of 'the ripple effect through entire supply-chain networks'.

**Response:** Thank you for your comments. In light of these comments, we have made further edits to highlight that we only consider upstream effects. Please see the yellow highlights in the manuscript.

[revised manuscript text omitted]